# Development of a PCR-based, genetic marker resource for the tomato-like nightshade relative, *Solanum lycopersicoides* using whole genome sequence analysis

Puneet Kaur Mangat, Ritchel B. Gannaban, Joshua J. Singleton, Rosalyn B. Angeles-Shim🄳 *

Department of Plant and Soil Science, College of Agricultural Sciences and Natural Resources, Texas Tech University, Lubbock, Texas, United States of America

* rosalyn.shim@ttu.edu

**Data Availability Statement:** All relevant data are within the manuscript and its Supporting Information files.

## Abstract

*Solanum lycopersicoides* is a wild nightshade relative of tomato with known resistance to a wide range of pests and pathogens, as well as tolerance to cold, drought and salt stress. To effectively utilize *S. lycopersicoides* as a genetic resource in breeding for tomato improvement, the underlying basis of observable traits in the species needs to be understood. Molecular markers are important tools that can unlock the genetic underpinnings of phenotypic variation in wild crop relatives. Unfortunately, DNA markers that are specific to *S. lycopersicoides* are limited in number, distribution and polymorphism rate. In this study, we developed a suite of *S. lycopersicoides*-specific SSR and indel markers by sequencing, building and analyzing a draft assembly of the wild nightshade genome. Mapping of a total of 1.45 Gb of *S. lycopersicoides* contigs against the tomato reference genome assembled a moderate number of contiguous reads into longer scaffolds. Interrogation of the obtained draft yielded SSR information for more than 55,000 loci in *S. lycopersicoides* for which more than 35,000 primers pairs were designed. Additionally, indel markers were developed based on sequence alignments between *S. lycopersicoides* and tomato. Synthesis and experimental validation of 345 primer sets resulted in the amplification of single and multilocus targets in *S. lycopersicoides* and polymorphic loci between *S. lycopersicoides* and tomato. Cross-species amplification of the 345 markers in tomato, eggplant, silverleaf nightshade and pepper resulted in varying degrees of transferability that ranged from 55 to 83%. The markers reported in this study significantly expands the genetic marker resource for *S. lycopersicoides*, as well as for related *Solanum spp.* for applications in genetics and breeding studies.

## Introduction

*Solanum lycopersicoides* ($2n = 2x = 24$) from the *Lycopersicoides* section of the genus *Solanum* is a wild nightshade species that is distantly related to the cultivated tomato (*S. lycopersicum*)

**Funding:** The author(s) received no specific funding for this work.

**Competing interests:** The authors have declared that no competing interests exist.

[1]. It is endemic to the west Andes by the Chile-Peru border and thrives at high altitudes of up to 3800 m above sea level [2]. The species has known adaptation to cold, drought and salt stress [3–5], as well as resistance to phytophagous pests (i.e. leafminers) [6] and pathogenic fungi (e.g. *Botrytis cinerea* and *Phytophthora parasitica*) [7,8], bacteria (e.g. *Xanthomonas campestris*, *Clavibacter michiganensis* subsp. *michiganensis* and *Pseudomonas syringae* pv. *tomato*) [6,9] and viruses (e.g. tomato mosaic virus, tomato yellow leaf curl virus and tomato crinivirus) that commonly afflict the cultivated tomato [10–12].

Early efforts to introgress desirable traits from *S. lycopersicoides* to tomato have led to the successful generation of diploid $F_1$ hybrids via embryo rescue following wide hybridization. Unfortunately, the resulting $F_1$s are functionally male-sterile and unilaterally incompatible with tomato pollen and hence cannot be used directly for backcrossing [13]. Despite the initial challenges, introgression of *S. lycopersicoides* chromosomes in the genetic background of tomato was achieved through various strategies developed to overcome reproductive barriers associated with the interspecific crossing. These included the combined use of male-fertile amphidiploids from the interspecific $F_1$ hybrids and bridging lines of *S. pennellii* to circumvent the issue of unilateral incompatibility [14], modification of bud pollination to systematically avoid or suppress crossability barriers [8], and synthesis of a partially male-fertile $F_1$ hybrid by pollinating tomato with pollen pooled from several *S. lycopersicoides* plants [6]. Adoption of these techniques led to the successful generation of *S. lycopersicoides*-derived monosomic alien addition lines (MAALs) and chromosome segment substitution lines (CSSLs) [6,14,15]. MAALs are plants having the full chromosome complement of the cultivated species used as the recipient parent in an interspecific cross and an extra chromosome from the wild relative donor (2n + 1) [16]. CSSLs on the other hand, comprise a set of plants in the genetic background of an elite cultivar that represent the whole genome of the wild species in small, contiguous or overlapping chromosome segments [17–19]. These pre-breeding materials are unique genetic resources that capture novel genetic variations from the wild nightshade species in the tomato background. To date, these lines have been extensively evaluated for a range of agronomic characteristics [7,20] but the limited availability of DNA-based markers that can unlock the genetic basis of the observed phenotypes has restricted their efficient utilization in actual breeding programs to improve tomato.

Marker systems that have been used to characterize pre-breeding materials derived from *S. lycopersicoides* include morphological, biochemical and molecular markers. The DNA markers are in the form of restriction fragment length polymorphism (RFLP) and simple sequence repeats (SSRs) that are based solely on the tomato genome [6,14,15,21]. More recently, PCR-based, cleaved amplified polymorphic sequence (CAPS) markers developed based on existing RFLPs have also been used to map chromosome introgressions from *S. lycopersicoides* in the tomato background [22]. Despite the availability of DNA markers to characterize pre-breeding stocks developed from *S. lycopersicoides*, the marker resource available for the species remains limited in number, genome coverage and polymorphism rate. In case of the RFLP and CAP markers, digestion reactions that add to the cost, time and labor necessary to complete the genotyping make them less ideal for genetics and breeding studies.

Advances in molecular biology and instrumentation have facilitated the development of next generation, technological platforms for the rapid sequencing and assembly of whole genomes of several species [23,24]. With the availability of sophisticated but user-friendly computational tools, interrogation of new genome assemblies for sequence variations such as SSRs, insertions/deletions (indels) and single nucleotide polymorphism (SNPs) that can be used as targets for molecular marker development has become mainstream [25–27].

SSRs are tandemly arranged, repetitive sequences that make up a significant portion of eukaryote genomes [28], whereas indels are genomic insertions and deletions resulting from

replication slippage, simple sequence replications, unequal crossovers, retrotransposon insertions and segmental duplications [29–31]. Markers based on SSRs and indels are co-dominant, highly polymorphic and abundant in the genome. They are easily assayed by PCR and the amplicons directly resolved by agarose gel electrophoresis without any additional steps, making them more economical [32–34]. The robustness and technical simplicity of these markers for routine genotyping make them the marker of choice for genetics and breeding applications in many laboratories.

In this study, we aim to expand the limited genetic marker resource for *S. lycopersicoides* by developing SSR and indel markers based on whole genome sequence analysis. The molecular markers generated in this study are expected to accelerate basic research on the discovery and functional validation of genes/quantitative trait loci (QTL) conditioning traits of agronomic value in S. *lycopersicoides* towards their utilization in breeding for trait improvement in tomato.

## Materials and methods

### Plant materials, DNA extraction and whole genome sequencing

Seeds of the wild nightshade species, *S. lycopersicoides* (Acc LA1964) were provided by the Tomato Genetics Resource Center of the University of California, Davis (http://tgrc.ucdavis. edu). Seeds were surface-sterilized with 1% hypochlorite solution, plated in petri dishes lined with moist, sterile paper towels and germinated at an ambient temperature of 25˚C in the laboratory. The germinated seeds were transferred individually in 1-litre pots containing conventional potting media (composed of 45–50% composted pine bark, vermiculite, Canadian sphagnum, peat moss, perlite and dolomitic limestone) supplemented with slow-release NPK fertilizers and maintained in the greenhouse of the Horticultural Gardens of the Department of Plant and Soil Science (PSS) at Texas Tech University. Total genomic DNA was isolated from young leaves of *S. lycopersicoides* following a modified CTAB method [35]. The quality and quantity of purified genomic DNA were estimated using the NanoDrop™ One Microvolume UV-Vis Spectrophotometer (ThermoFisher, USA). Library preparation and whole genome sequencing using the Illumina HiSeq 3000 PE150 platform was outsourced to the Clinical Genomics Center of the Oklahoma Medical Research Foundation.

### Whole genome analysis for repetitive sequence detection and primer design

Raw sequence data composed of 151-bp paired end reads were filtered using the Trimmomatic tool [36] to remove the Illumina adapter sequences and reads with poor quality. The trimmed reads were then assessed for quality using FastQC [37] before *de novo* assembling them into contigs using the short-read assembler, ABySS 2.0 [38]. Guided by the Build 3.0 of the reference genome for tomato (cv. Heinz), the contigs were used to generate longer scaffolds using the post-assembly genome improvement toolkit or PAGIT [39]. The quality of the newly built, draft assembly in comparison to the tomato reference genome was assessed using the quality assessment tool for genome assemblies or QUAST [40].

Genomic features in the draft assembly that can be used as basis for primer design were identified using various computational tools. Families of repetitive sequences were determined *de novo* based on existing Repbase libraries and collated into a species-specific database using RepeatModeler [41]. Repbase is maintained by the Genetic Information Resource Institute (GIRI) and is used as a reference for the annotation of eukaryotic repetitive DNA [42]. Repbase has libraries that are specifically available for the RepeatMasker software. The identified repeats were classified and annotated as retrotransposons, DNA transposons, small RNAs, satellites, simple repeats or low complexity DNA sequences using the RepeatMasker program

[43]. Mining the assembly for SSRs was carried out using the GMATA software [44]. SNPs and indels were identified based on sequence alignment between the *S. lycopersicoides* assembly and the tomato reference genome using the NUCmer (nucleotide MUMmer) package of the MUMmer program [45]. All programs used in the study ran on default settings.

The Primer3 program integrated into the GMATA software was used to generate primers for the SSRs that were identified in the draft assembly. To design the indel markers, comparative sequence alignment between the draft *S. lycopersicoides* and reference tomato genomes was generated using the Burrows-Wheeler aligner (BWA) [46]. The output file was then converted into a binary file that can be viewed using the Integrative Genomics Viewer [47]. Primers targeting the indels were manually designed using open source software and following specifications for standard primer design (i.e. 40–60% GC content, 20–25 bp in length). The GC content of the primer sequences were validated using the EndMemo-DNA/RNA GC content calculator [48]. The reverse primers were generated by reverse complementing DNA sequences through http://arep.med.harvard.edu/labgc/adnan/projects/Utilities/revcomp.html [49].

All SSRs and indel primers were designed at an average chromosome interval of 2–2.6 Mb. The specificity of the designed primers was validated *in silico* by BLAST searches [50] against the available tomato sequences curated at the NCBI database. Synthesis of all SSR and indel primers was outsourced to Sigma, USA.

### Target amplification and cross-species transferability of *S. lycopersicoides* DNA markers

The ability of the newly designed markers to amplify targets in *S. lycopersicoides* was validated following a standard PCR protocol [51]. Adjustments in annealing temperature from 53˚C to 55˚C were carried out to optimize target amplification in *S. lycopersicoides*. PCR amplicons were resolved in 3% agarose gel in 1X Tris-Borate-EDTA buffer [51].

Additionally, the transferability of the *S. lycopersicoides*-specific markers to two other *S. lycopersicoides* accessions (LA2951 and LA2387) and other Solanaceous plants including tomato (Acc LA3122), eggplant (*S. melongena*) cv. Black Beauty, pepper (*Capsicum annuum*) cv. California Wonder and silverleaf nightshade (*S. elaeagnifolium*) was also determined. Young leaves from tomato, eggplant and pepper were sampled from seedlings germinated in the greenhouse as previously described. Leaf tissues of silverleaf nightshade were randomly sampled from populations growing at the Horticultural Gardens of PSS. Total genomic DNA for PCR was extracted from the young leaves of each of the Solanaceous species following a modified CTAB method [35].

## Results and discussion

### Whole-genome assembly and sequence repeats analysis

Illumina sequencing generated a total of 15.8 Gb of raw data containing 88,457,926 paired end reads that are 151 bp long (SRA accession SRX9292807). After trimming the adapters and removal of the poor-quality reads, the calculated average genome coverage [52] based on the tomato reference genome was 25X. *De novo* assembly generated a total of 6,874,225 contigs spanning a total length of 1,452,602,585 bp (Table 1).

Reference-guided assembly of the contigs into longer scaffolds mapped 589,717,391 bp (40.59%) of the *S. lycopersicoides* sequence data against the tomato genome. *S. lycopersicoides* and tomato are distant relatives that belong to different sections under the genus *Solanum*. The former belongs to the section *Lycopersicoides* which also includes one other species, *S.*

**Table 1. General statistics obtained for *S. lycopersicoides* genome assembly using AbySS.**

| Descriptive statistics | Value (bp) |
| --- | --- |
| Number of contigs | 6,874,225 |
| Total length | 1,452,602,585 |
| Largest contig size | 46755 |
| Number of contigs that are ≥500bp | 394658 |
| Reference length[a] | 828,076,956 |
| Total length of contigs aligned to the reference | 589,717,391 |
| N50[b] | 2141 |

[a]length of the Build 3.0 of tomato cv. Heinz reference genome.

[b]N50 is the length for which the collection of all contigs of that length or longer covers at least 50% of the assembly.

*sitiens*, whereas the latter belongs to the section *Lycopersicon* which includes twelve other wild relatives [53]. Throughout the course of evolution, the genomes of these plants have been subjected to mutations, chromosomal rearrangements, transposon amplifications, gene duplications and extensive genome expansion/contraction. The genetic differentiation of each species that resulted from such genomic events may explain the moderate alignment of the *S. lycopersicoides* contigs against the tomato reference genome. This observation is consistent with the high proportion (17–25%) of paired sequence reads generated for *S. arcanum*, *S. pennellii* and *S. habrochaites* that also did not map against Build 2.40 of the tomato cv. Heinz reference genome, despite the three species belonging to the same section as tomato [54]. Given the 1.45 Gb total contig length obtained for *S. lycopersicoides* in this study, assembly of a draft that is guided by a genome of a closer relative other than tomato has the potential to generate a longer consensus sequence for the species. Alternatively, the draft assembly can be improved by refining, gap filling and expanding the initial assembly with long reads generated by third-generation sequencing technology such as the PacBio SMRT.

Interrogation of the draft assembly for repetitive sequences using the RepeatModeler in conjunction with the RepeatMasker detected a total of 712,011 repeats, covering 164,390,084 bp (18.83%) of the draft. The interspersed repeats consisted of short interspersed nuclear elements (0.06%), short interspersed nuclear elements (0.85%), long terminal repeats (6.63%), DNA elements (1.24%) and unclassified repeats (9.63%). The proliferation and deletion of transposable elements (TEs) are key determinants of genome size variation in eukaryotes [55]. The loss of TEs in tomato during domestication may be one of the primary reasons behind the genome size difference between *S. lycopersicoides* and tomato. RepeatMasker also classified the repeats into small RNAs (0.05%), satellites (0.01%), simple repeats (0.39%) and low complexity repeats (0.08%) (Fig 1). With the draft assembly capturing less than 50% of the *S. lycopersicoides* genome, analysis of an improved assembly is expected to increase the proportion of these repeats in the genome.

SSR mining identified 56,901 SSRs with motifs ranging from 2 to 9 bp (Table 2). SSRs with di-nucleotide motifs were the most abundant (74.05%), whereas those having penta-, hexa-, hepta-, octo- and nona-nucleotide repeats comprise only approximately 1.26% of the total SSRs identified in the assembly. Among the di-nucleotide motifs, AT and TA make up more than 50% of the total SSRs with 2-bp repeats (Fig 2A). In terms of length, 10-bp SSRs were the most predominant (37.9%), whereas those that are 34-bp long were the scarcest (0.5%) (Fig 2B).

## Primer design and target DNA amplification

The built-in Primer3 software in the GMATA tool was used to design primer pairs for 35,801 SSR loci with 34,198 unique markers (Table 3). To validate the ability of primer pairs to

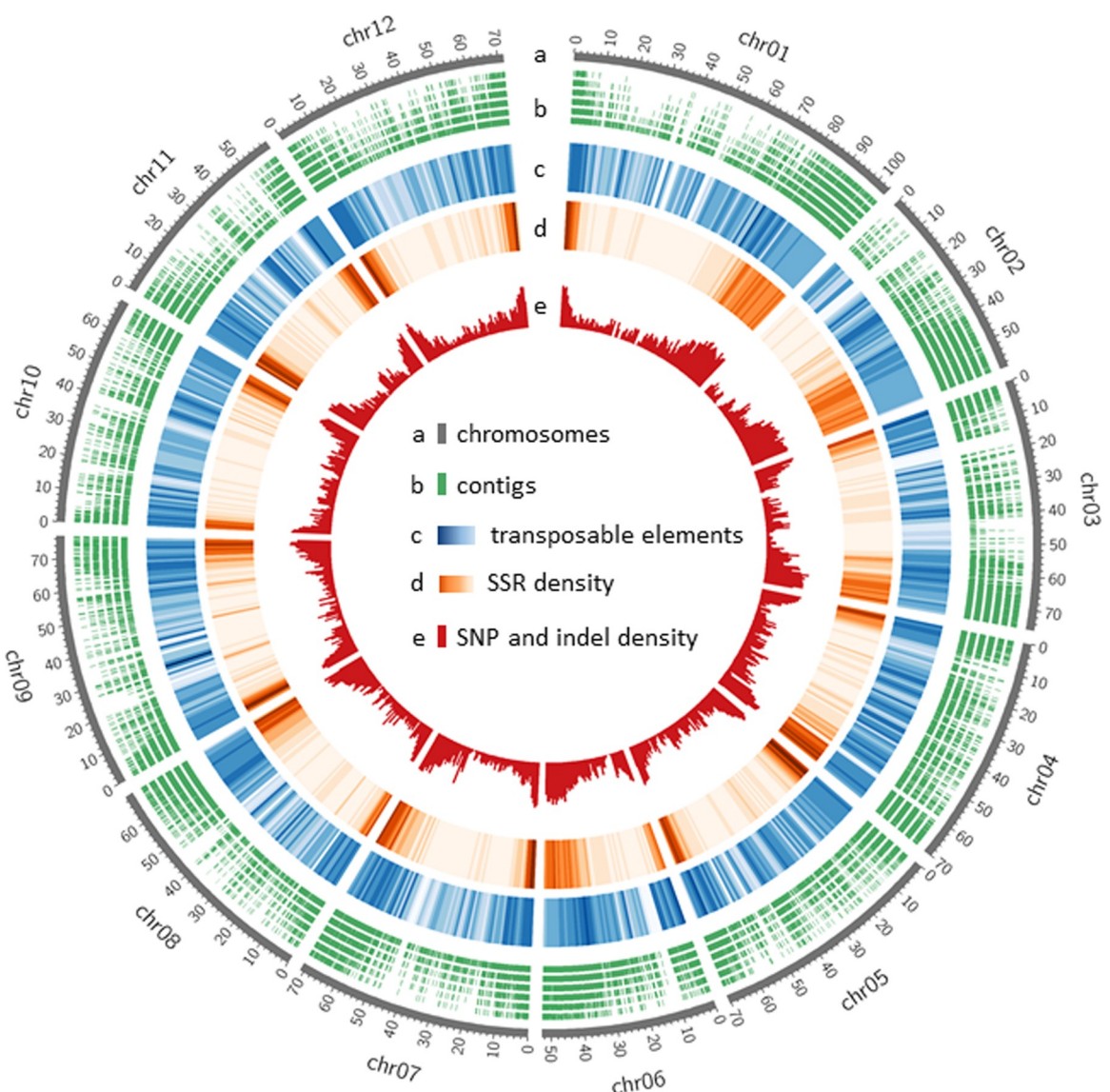

**Fig 1. Circular view of the *S. lycopersicoides* genome assembly used for sequence variation mining.** The outermost to the innermost rings represent the 12 representative pseudomolecules, contigs (≥500 bp), transposable elements, SSRs, and indels and SNPs. Color keys for the transposable elements and SSRs indicate the density of the repeats. The more intense the color, the more repetitive sequences in the pseudomolecule position. The indel and SNP density was determined based on sequence alignments between *S. lycopersicoides* and tomato. All tracks show binned data with a window size of 1 Mb.

amplify targets in *S. lycopersicoides*, primers targeting 196 SSRs with di-nucleotide to hexa-nucleotide repeat motifs were selected (S1 Table). All primer pairs were 20–25 bp long and have an estimated amplicon size of 150–350 bp. Of the 196 SSRs, 182 successfully amplified targets in *S. lycopersicoides* Acc LA1964, with 148 annealing at 55°C and two at 53°C (Tables 4 and S2). Fifty-nine of the SSRs were multilocus, with 33 amplifying two bands and 26 amplifying more than two bands.

In addition, 149 indel markers (Tables 4 and S1) were also manually designed based on the alignment between the *S. lycopersicoides* draft assembly and the available reference genome for tomato. Of the 149 primer pairs, 143 successfully amplified targets in *S. lycopersicoides*

**Table 2. SSRs mined from assembled *Solanum lycopersicoides* genome using GMATA software.**

| Motif(-mer) [a] | Total | Percentage (%) |
|---|---|---|
| 2 | 42,135 | 74.05 |
| 3 | 12,691 | 22.30 |
| 4 | 1,353 | 2.38 |
| 6 | 350 | 0.62 |
| 5 | 303 | 0.53 |
| 7 | 64 | 0.11 |
| 9 | 3 | 0.01 |
| 8 | 2 | 0.00 |

[a]range of motif length was chosen while running the software.

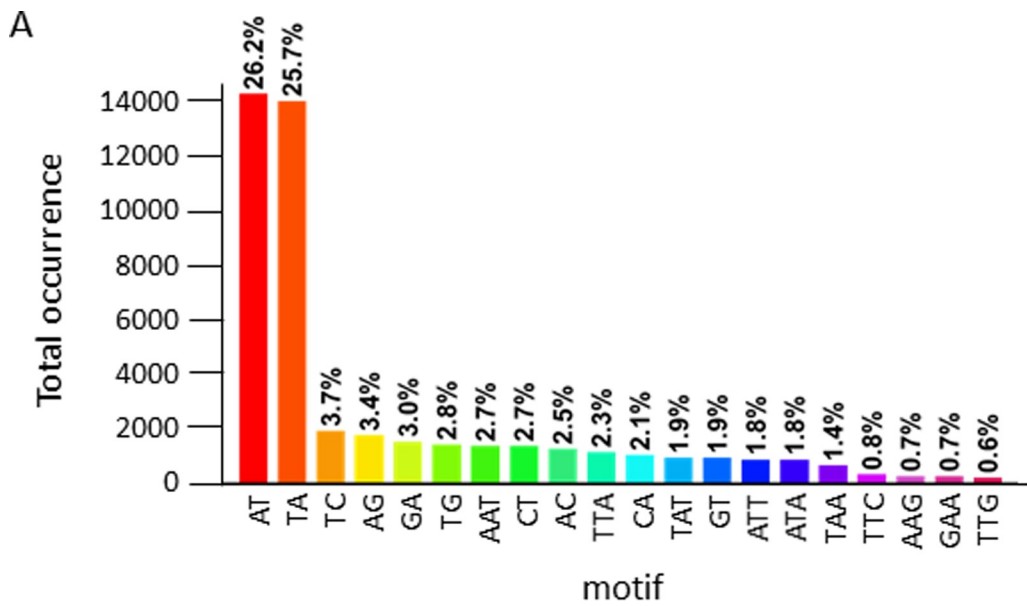

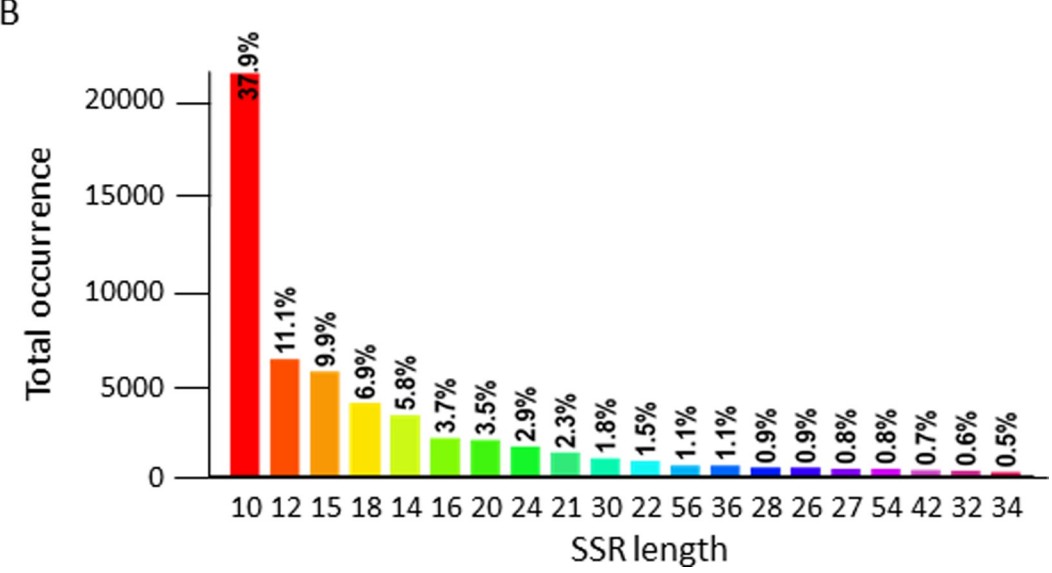

**Fig 2.** Distribution of the different (A) SSR motifs and (B) SSR lengths throughout the draft assembly of *S. lycopersicoides* based on GMATA data.

**Table 3. Summary of primer pairs designed by Primer3 of GMATA based on the SSRs mined.**

| | Total | Percentage (%) |
|---|---|---|
| total no. of loci detected | 56,901 | - |
| no. of SSR loci with designed primer pairs | 35,801 | 62.90 |
| no. of SSR loci without designed primer pairs | 21,100 | 37.00 |
| no. of unique markers | 34,198 | - |

genome, with 127 annealing at 55°C and 16 annealing at 53°C (S2 Table). Ten primer pairs amplified two bands, whereas two amplified more than two bands, indicating multilocus targets for the designed primers.

In summary, a total of 345 markers composed of 196 SSRs and 149 indels that are distributed across the 12 draft pseudomolecules of *S. lycopersicoides* were designed at an average map interval of 2.6 bp (Fig 3). Of these, 326 (94.50%) amplified targets in *S. lycopersicoides* Acc 1964. A slightly lower transferability of the markers was observed for *S. lycopersicoides* accessions LA2951 (68.00%) and LA2387 (70.00%), although multilocus amplifications were also observed. Of the total number of indels and SSRs tested, 11 and 24% amplified multiple targets in LA2951, and 27 and 49% in LA2387, respectively.

Previous studies on *S. lycopersicoides* have relied heavily on the use of RFLPs, SSRs and CAPS designed based on the tomato genome [15,21,56,57]. While these markers have been useful for genetic diversity studies and for monitoring wild chromosome introgressions, their distribution and number are not sufficient for mapping and cloning useful genes/QTLs in *S. lycopersicoides*. The newly designed markers in this study, 94% of which amplified their target loci, offers a much broader marker resource that can be used in genetics and breeding studies on *S. lycopersicoides*.

## Cross-species transferability of *S. lycopersicoides*-specific markers

Cross-species amplification of all 345 *S. lycopersicoides*-specific markers in tomato, eggplant, silverleaf nightshade and pepper resulted in varying degrees of transferability ranging from 55% to 83% (Table 4 and Fig 4). In tomato, 148 SSRs and 138 indels amplified, with 20 markers showing multilocus targets. A total of 206 (59.71%) SSRs and indels amplified polymorphic targets between *S. lycopersicoides* and tomato, indicating the potential of these markers in mapping genes/QTLs in pre-breeding materials derived from crosses between the two species.

After tomato, silverleaf nightshade recorded the most number of target loci for the markers followed by eggplant and pepper. Section *Lycopersicoides* to which *S. lycopersicoides* belongs is an immediate outgroup of the tomato clade [53], making *S. lycopersicoides* closest to tomato. In contrast, pepper, which belongs to a separate genus, is the most distant to *S. lycopersicoides*.

**Table 4. Target amplification and cross-species transferability of *S. lycopersicoides*-specific SSR and indel markers.**

| Plant species | No. of markers tested | | No. of markers that amplified | | Transferability rate (%) |
|---|---|---|---|---|---|
| | SSR | indel | SSR | indels | |
| *S. lycopersicoides* (Acc LA1964) | 196 | 149 | 182 | 143 | 94.20 |
| *S. lycopersicoides* (Acc LA2951) | 196 | 149 | 123 | 114 | 68.69 |
| *S. lycopersicoides* (Acc LA2387) | 196 | 149 | 124 | 119 | 70.43 |
| *S. lycopersicum* | 196 | 149 | 148 | 138 | 82.90 |
| *S. melongena* | 196 | 149 | 111 | 90 | 58.26 |
| *S. elaeagnifolium* | 196 | 149 | 148 | 115 | 76.23 |
| *C. annum* | 196 | 149 | 105 | 83 | 54.49 |

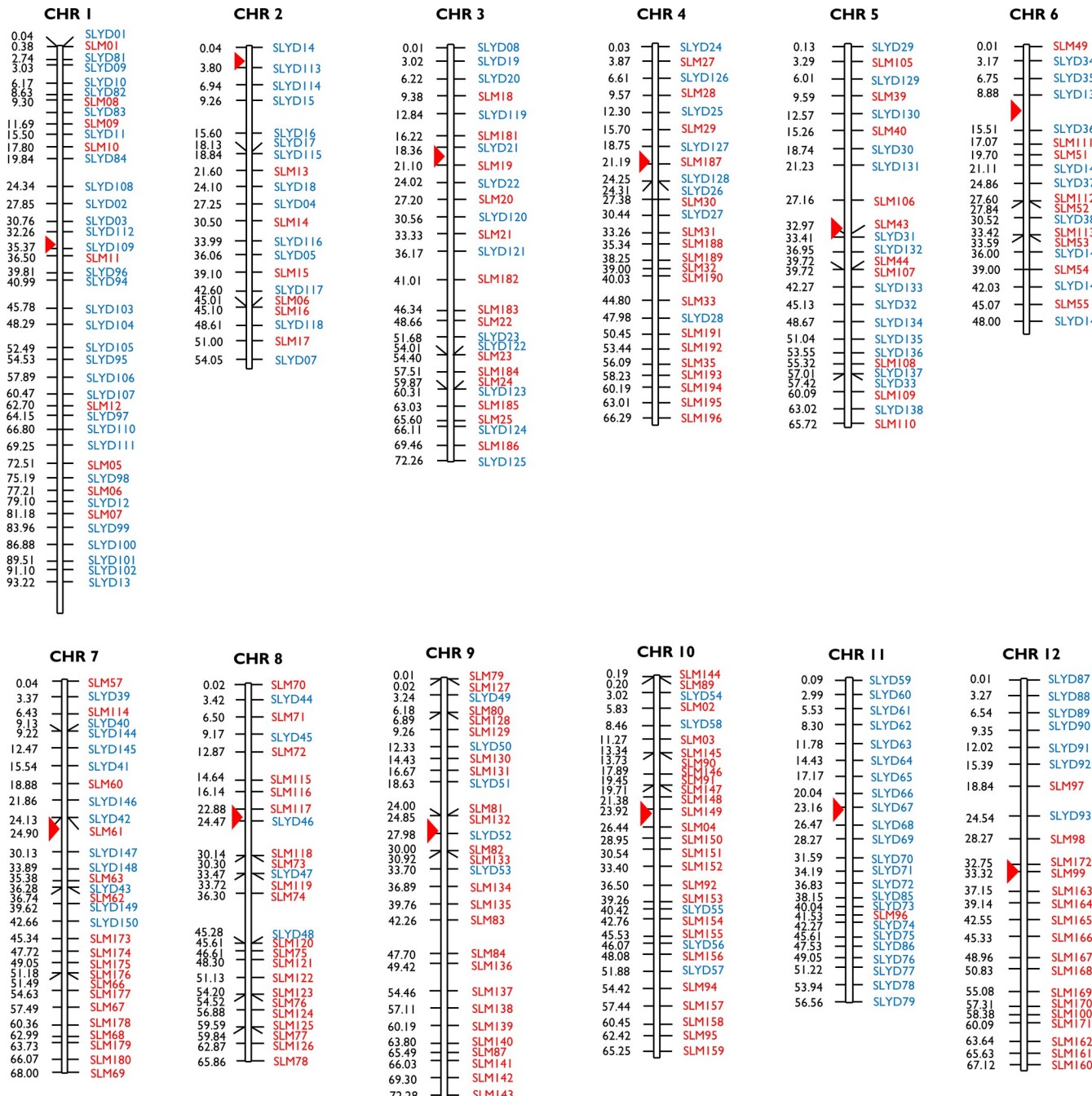

**Fig 3. Chromosome distribution of *S. lycopersicoides*-specific SSR and indel markers.** Map position of all the markers is based on their position in tomato chromosomes. Red markers are SSRs and blue markers are indels. Red triangle = centromere.

The degree of the genetic relatedness of *S. lycopersicoides* to either species is consistent with the highest and lowest rate of marker cross-transferability recorded for tomato and pepper, respectively. In a similar manner, the phylogenetic relationship of *S. lycopersicoides* to eggplant and silverleaf nightshade reflects the observed rate of marker transferability to the latter two species. Silverleaf nightshade and eggplant belong to the subgenus *Leptostemonum* of the genus *Solanum*. Compared to tomato, silverleaf nightshade and eggplant are more distantly related

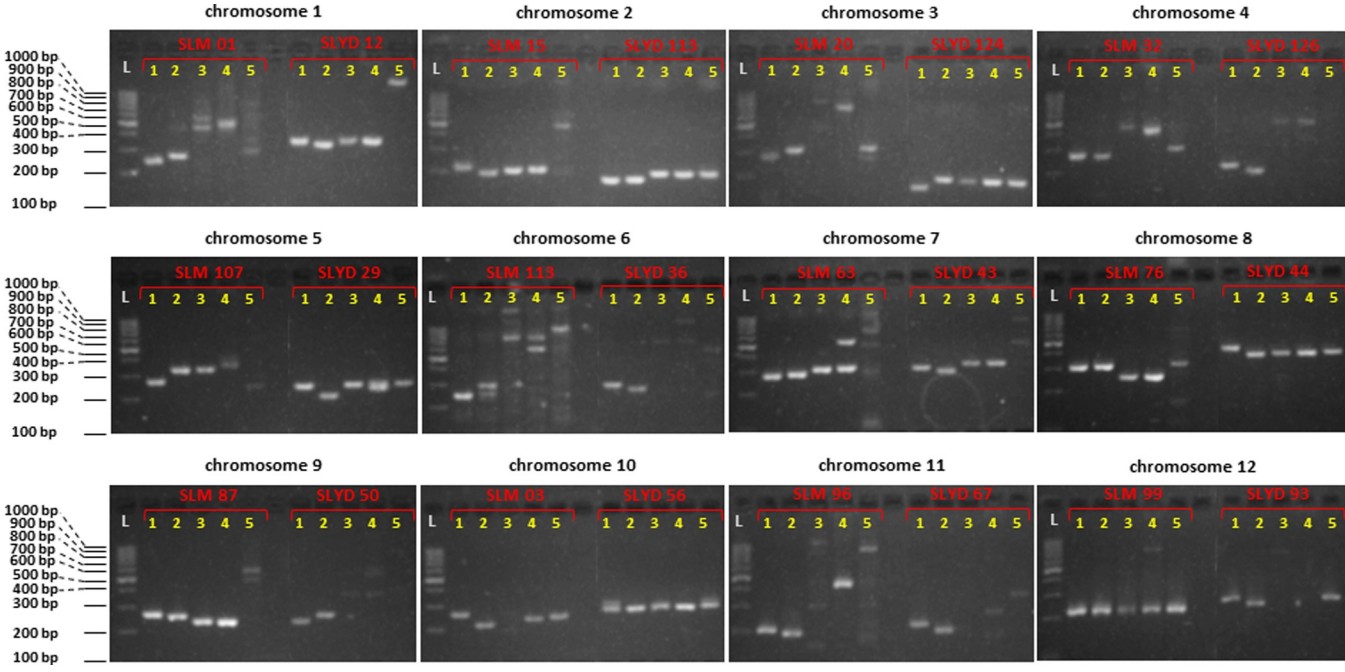

**Fig 4. Cross-species amplification of *S. lycopersicoides*-specific markers in other members of Solanaceae.** One SSR and one indel marker for each chromosome were used to amplify targets in tomato, silverleaf nightshade, eggplant and pepper. SLM = SSR marker, SLYD = indel marker, 1 = *S. lycopersicoides*, 2 = tomato, 3 = silverleaf nightshade, 4 = eggplant, 5 = pepper, L = 100 bp-ladder.

to *S. lycopersicoides* [58] hence the lower rate of marker transferability in these two species compared to tomato. Comparative genomic studies in eggplant, potato, pepper and tomato indicate the highly conserved linkage order for markers despite the occurrence of major inversion events that drove the evolution of these related genomes [59–61]. This further support the relatively high transferability of *S. lycopersicoides* markers in the closely related species.

The generally high rates of cross-species amplification of *S. lycopersicoides* markers indicate their potential use in genetics and breeding applications in related Solanaceous plants. In fact, a subset of 54 *S. lycopersicoides*-specific, indel markers have been successfully used to assess the genetic diversity in silverleaf nightshade populations from different localities in Texas, USA. Genetic profiling using the indels, along with other DNA markers from related species, established the genetic differentiation of silverleaf nightshade populations in response to variations in selection pressures that are unique to the ecological habitats selected in the study [62].

## Conclusions

Tomato production amidst worsening agro-environments can be sustained by harnessing natural genetic variation from wild tomato relatives that can provide durable forms of adaptation to the crop against both biotic and abiotic stresses. *S. lycopersicoides* is a distant tomato relative with known adaptation to marginal environments. Exploiting the genetic potential of *S. lycopersicoides* for tomato breeding will require understanding of the genetic basis of the adaptability of this wild species.

We designed and validated a total of 345 SSR and indel markers that are specific to *S. lycopersicoides* using whole genome sequence analysis. These markers, together with the more than 30,000 SSRs that are available for validation significantly expands the genetic marker resource that can be used for QTL analysis, mapping and positional cloning of genes in *S.*

*lycopersicoides* that can be utilized towards value-added trait improvements in tomato. The transferability of the *S. lycopersicoides* markers to tomato, eggplant, pepper and silverleaf nightshade indicate their applicability in similar genetics and breeding studies in these Solanaceous species.

## Supporting information

**S1 Table. *S. lycopersicoides*-specific DNA markers developed using whole genome sequence analysis.**
(DOCX)

**S2 Table. Forward and reverse primer sequences of *S. lycopersicoides*-specific markers.**
(DOCX)

**S1 Raw images. Amplification of *S. lycopersicoides*-specific markers in other Solanaceous species.** One SSR and one indel marker for each chromosome were used to amplify targets in tomato, silverleaf nightshade, eggplant and pepper. SLM = SSR marker, SLYD = indel marker, 1 = *S. lycopersicoides*, 2 = tomato, 3 = silverleaf nightshade, 4 = eggplant, 5 = pepper, L = 100 bp-ladder. Lanes marked in X were not used to generate Fig 4.
(PDF)

## Acknowledgments

The authors would like to thank the Tomato Genetics Resource Center (TGRC) of the University of California, Davis for generously providing us with the seeds of tomato Acc LA3122, and *S. lycopersicoides* accessions LA1964, LA2951 and LA2387.

## Author Contributions

**Conceptualization:** Rosalyn B. Angeles-Shim.

**Data curation:** Puneet Kaur Mangat, Joshua J. Singleton.

**Formal analysis:** Puneet Kaur Mangat, Ritchel B. Gannaban.

**Writing – original draft:** Puneet Kaur Mangat, Rosalyn B. Angeles-Shim.

**Writing – review & editing:** Puneet Kaur Mangat, Rosalyn B. Angeles-Shim.

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
