## [Decision Letter · Decision Letter 0]

18 Sep 2020

PONE-D-20-20252

Development of PCR-based, genetic marker resource for the tomato-like nightshade relative, Solanum lycopersicoides using whole genome sequence analysis

PLOS ONE

Dear Dr. Angeles-Shim,

Thank you for submitting your manuscript to PLOS ONE. After careful consideration, we feel that it has merit but does not fully meet PLOS ONE’s publication criteria as it currently stands. Therefore, we invite you to submit a revised version of the manuscript that addresses the points raised during the review process.

The key point is to deposit your sequences and any related information into a public database so that readers can use them later. In your revised manuscript, the accession # of your deposits should be listed. 

We look forward to receiving your revised manuscript.

Kind regards,

David D Fang, Ph.D.

Academic Editor

PLOS ONE

Journal Requirements:

Reviewers' comments:

Reviewer's Responses to Questions

**Comments to the Author**

1. Is the manuscript technically sound, and do the data support the conclusions?

Reviewer #1: Partly

Reviewer #2: Yes

Reviewer #3: Yes

2. Has the statistical analysis been performed appropriately and rigorously? 

Reviewer #1: Yes

Reviewer #2: Yes

Reviewer #3: N/A

3. Have the authors made all data underlying the findings in their manuscript fully available?

Reviewer #1: No

Reviewer #2: Yes

Reviewer #3: No

4. Is the manuscript presented in an intelligible fashion and written in standard English?

Reviewer #1: Yes

Reviewer #2: Yes

Reviewer #3: Yes

5. Review Comments to the Author

Reviewer #1: The main interest of the ms is the data obtained to help to tomato breeders. But most of the generated data are not available; such as raw sequences, assembled draft sequences, marker complete sequences. All these sequences have to be deposited on public repositories, NCBI or as supplemental data.

The authors show only the localization and primer sequences of the 345 markers tested; but not of the all identified markers.

An author has been already published 54 markers of 345 in other ms.

The authors identify SSR and indels, but a SNP calling with standard software will be very useful to identify indels and SNPs between tomato ans S. lycopersicoides.

Also test some SSR markers and indels in several accesions of S. lycopersicoides could allow to researchers to identify really polymorphic markers in this species.

Reviewer #2: In this manuscript entitled ‘Development of PCR-based, genetic marker resource for the tomato-like nightshade relative, Solanum lycopersicoides using whole genome sequence analysis’, the authors presented whole genome sequencing for a Solanum lycopersicoides accession and identified a large number of SSRs and indels in comparison to other tomato reference genomes. The primer sets were produced and evaluated their PCR amplification and polymorphism between Solano lycopersicoides and tomato. In addition cross-species transferability of these Solanum lycopersicoides-specific markers was assessed using Solanaceeous plants. The manuscript was well written and methods for genome sequencing and marker development were clearly described. I think this manuscriot is acceptable for publication in this journal. The SSR and indel marker information provided in this paper will be useful for genetic mapping and other related studies. Some of my stand points for revision is as following:

In line 85, ‘cleaved amplified polymorphic sequence (CAPS) marker’ is more common name

In Table 4, please describe the polymorphism rate between S. lycopersicoides and other species

Please, provide a figure showing gel image of marker transferability.

Please, provide gene annotation information for each markers

Finally, conclusion is lengthy and overlaps with the introduction for some parts. Please, make it concise by focusing on the results and its importance.

Reviewer #3: This work provides a nice resource for those interesting in studying characteristics of lycopersicoides. Additional comments:

Line 60: Also resistance to Pseudomonas, cite appropriate paper

Line 112: Why did you choose this accession?

Line 119: Did you pool plants or choose a single individual for sequencing? A pooled sample could partially explain the poor assembly

Line 132: S. lycopersicoides has some structural variation relative to Heinz, so you probably missed this by using Heinz as a scaffolding reference.

Line 135: Which “existing libraries” did you use for repeat detection?

Line 143: I don’t understand what is going on here. You already generated an alignment using mummer, so why generate it again with BWA? I would think the mapping rate would be lower here too, but I don’t see that result.

Line 146: What “open source” software did you use?

You should check that your primers are specific in lycopersicoides by sequence analysis. This may have reduced the number of multilocus amplifications you got with your PCRs.

Options for all programs used should be included in the methods. It would be very hard to repeat this work based on the details given.

Line 171: Your sequencing coverage is pretty low for de novo assembly

Line 178: You might have been able to adjust your mapping parameters to get better alignment, but since you don’t give these parameters it is hard to know.

Line 190: There is a PacBio assembly of LA2951 already (https://www.biorxiv.org/content/10.1101/2020.04.16.039636v1.full)

Line 194: In closely related species, a lot more of the genome is annotated as repetitive. Why do you think your estimate is so low?

Line 201: I don’t understand why you say the assembly only captures less that 50% of the lycopersicoides genome. Your assembly length is actually a bit longer than the expected genome size. Are you basing this off of the % that mapped to Heinz? You should do Kmer analysis to see what the expected size is.

Line 231: Why would you do this manually? You have the mummer alignment and can generate a pipeline to do this.

Line 253: I’m curious why you did not use bioinformatics approaches to find the primers in these other species first. This approach would have given you a more refined list for PCR testing.

Figure1 is very low resolution in my copy

Figure 3: the label for chromosome 4 has shifted to a different line

Do the markers work in other lycopersicoides accessions? You could check LA2951 which is a more finished assembly.

How did you use BWA to align Lyd to Heinz? Lind 144

How did you handle multiply mapped contigs or regions of the Heinz genome that had many contigs aligning?

The reads should be submitted to SRA and the markers to solgenomics.net

6. PLOS authors have the option to publish the peer review history of their article (what does this mean?). If published, this will include your full peer review and any attached files.

Reviewer #1: No

Reviewer #2: No

Reviewer #3: No

---

## [Author Response · Author response to Decision Letter 0]

4 Nov 2020

Response to Reviewers’ comments

Reviewer #1

Comment 1: The main interest of the ms is the data obtained to help to tomato breeders. But most of the generated data are not available; such as raw sequences, assembled draft sequences, marker complete sequences. All these sequences have to be deposited on public repositories, NCBI or as supplemental data.

Answer 1: Sequence information and physical location relative to the tomato genome of the 345 markers reported in this study is provided in Supplementary Tables 1 and 2. The raw sequences have been submitted to the SRA database of NCBI and assigned the accession number SRX9292807¬¬¬¬¬¬¬. 

Comment 2: The authors show only the localization and primer sequences of the 345 markers tested; but not of the all identified markers.

Answer 2: The Primer3 tool of GMATA software generated more than 35,000 primer pairs for the identified SSR loci. For the purpose of this study, we only selected 196 SSRs in addition to 149 indels markers that are distributed at an average interval of 2.6 bp for validation.

Comment 3: An author has been already published 54 markers of 345 in other ms.

Answer 3: Yes, a subset of 54 indel markers designed in this study have been used in the genetic diversity analysis of silverleaf nightshade from localities in Texas and this has been cited in the text as [62] (Line 371). Full information for the citation is provided in the references as “Singleton JJ, Mangat PK, Shim, J., Vavra C, Coldren C, Angeles-Shim RB. Cross-species transferability of Solanum spp. DNA markers and their application in assessing genetic variation in silverleaf nightshade (Solanum elaeagnifolium) populations from Texas, USA. Weed Sci. 2020:1-25. doi:10.1017/wsc.2020.25”. The detailed description of how this marker subset was designed and tested for its ability to amplify targets in S. lycopersicoides and other related Solanaceous species are reported for the first time here.

Comment 4: The authors identify SSR and indels, but a SNP calling with standard software will be very useful to identify indels and SNPs between tomato and S. lycopersicoides.

Answer 4: SNPs and indels were also identified based on sequence alignment between the S. lycopersicoides assembly and the tomato reference genome using the MUMmer program (Line 145-147). A summary of the distribution of the SNPs is presented in Fig. 1.

Comment 5: Also test some SSR markers and indels in several accessions of S. lycopersicoides could allow to researchers to identify really polymorphic markers in this species.

Answer 5: As suggested, the applicability of the 345 SSR and indel markers in other accessions of S. lycopersicoides was tested. Due to the poor viability of S. lycopersicoides Acc LA4131 and given the time constraints, the markers were tested on two other accessions only (i.e. LA2951 and LA2387). Results of the ability of the markers to amplify targets in other S. lycopersicoides accessions are presented in Table 4 and lines 168-169 and 256-259. 

Reviewer #2: In this manuscript entitled ‘Development of PCR-based, genetic marker resource for the tomato-like nightshade relative, Solanum lycopersicoides using whole genome sequence analysis’, the authors presented whole genome sequencing for a Solanum lycopersicoides accession and identified a large number of SSRs and indels in comparison to other tomato reference genomes. The primer sets were produced and evaluated their PCR amplification and polymorphism between Solanum lycopersicoides and tomato. In addition cross-species transferability of these Solanum lycopersicoides-specific markers was assessed using Solanaceous plants. The manuscript was well written and methods for genome sequencing and marker development were clearly described. I think this manuscript is acceptable for publication in this journal. The SSR and indel marker information provided in this paper will be useful for genetic mapping and other related studies. Some of my stand points for revision is as following:

Comment 1: In line 85, ‘cleaved amplified polymorphic sequence (CAPS) marker’ is more common name.

Answer 1: Recommended change has been made (Line 86).

Comment 2: In Table 4, please describe the polymorphism rate between S. lycopersicoides and other species.

Answer 2: The overall polymorphism rate of the markers between S. lycopersicoides and tomato is 59.71% (Line 287-289). Because the premise of this study is to allow the more efficient utilization of wild Solanum relatives in breeding for tomato, we purposely screened marker polymorphism between the two and but not between S. lycopersicoides and the other Solanaceous crops. Screening for transferability of the markers to other Solanaceous crop was a peripheral activity to aid other researchers/scientists needing more markers in their respective studies.

Comment 3: Please, provide a figure showing gel image of marker transferability.

Answer 3: Gel image showing marker transferability has been added (Figure 4).

Comment 4: Please, provide gene annotation information for each markers.

Answer 4: The chromosome location and map position of the markers relative to the tomato genome are provided in S1 Table along with other information such as repeat motif, expected amplicon and annealing temperature (S2 Table). Given the origin of SSRs from retrotransposon/transposable elements, the likelihood that the SSRs identified in the study are located in genic regions is low. Nevertheless, the marker information supplied with the manuscript can be used to validate if the SSR are within genic regions or not. This however, is beyond the scope of the study. 

Comment 5: Conclusion is lengthy and overlaps with the introduction for some parts. Please, make it concise by focusing on the results and its importance.

Answer 5: The conclusion was revised as suggested.

Reviewer #3: This work provides a nice resource for those interesting in studying characteristics of lycopersicoides. Additional comments:

Comment 1: Line 60: Also resistance to Pseudomonas, cite appropriate paper 

Answer 1: The suggested reference has been added as [9] (Line 61). Full information for the citation is provided in the references as “Mazo-Molina C, Mainiero S, Hind SR, Kraus CM, Vachev M, Maviane-Macia F, et al. The Ptr1 locus of Solanum lycopersicoides confers resistance to race 1 strains of Pseudomonas syringae pv. tomato and to Ralstonia pseudosolanacearum by recognizing the type III effectors AvrRpt2 and RipBN. Mol Plant Microbe Interact. 2019 Aug;32(8), 949-60. pmid:30785360”.

Comment 2: Line 112: Why did you choose this accession?

Answer 2: This study is part of a bigger project that focuses on the phenotypic and genotypic characterization of introgression lines derived from monosomic alien additional lines of S. lycopersicoides Acc. LA1964 in the background of cultivated tomato. Tomato markers have very low transferability to S. lycopersicoides LA1964 and we needed to develop a core set of S. lycopersicoides-specific markers that we can use to genotype the introgression lines.

Comment 3: Line 119: Did you pool plants or choose a single individual for sequencing? A pooled sample could partially explain the poor assembly

Answer 3: Only a single plant was sampled for genome sequencing. The assembly was reference-guided and we used the tomato genome as a reference. The genetic distance between the two species may also partially explain the low assembly.

Comment 4: Line 132: S. lycopersicoides has some structural variation relative to Heinz, so you probably missed this by using Heinz as a scaffolding reference.

Answer 4: Given the genetic distance between tomato and S. lycopersicoides, this is very likely as evident from the moderate number of contigs that we were able to map into scaffolds (Line 138). We discussed this in Lines 189-199.

Comment 5: Line 135: Which “existing libraries” did you use for repeat detection?

Answer 5: Repbase libraries were used for repeat detection [42]. A description of this library is provided in Lines 140-142.

Comment 6: Line 143: I don’t understand what is going on here. You already generated an alignment using mummer, so why generate it again with BWA? I would think the mapping rate would be lower here too, but I don’t see that result.

Answer 6: BWA alignment was generated initially between the tomato reference genome and S. lycopersicoides ABySS contigs to view the alignment in IGV. This was done to manually develop indel markers that we need for a bigger project mentioned in the response to Comment 2. The MUMmer alignment was performed later on using the draft assemblies.

Comment 7: Line 146: What “open source” software did you use?

Answer 7: EndMemo-DNA/RNA GC content calculator was used to check the GC content of the primer sequence [48]. To generate reverse primer, DNA sequence was reverse complemented using http://arep.med.harvard.edu/labgc/adnan/projects/Utilities/revcomp.html [49]. (Line 153-156) 

Comment 8: You should check that your primers are specific in lycopersicoides by sequence analysis. This may have reduced the number of multilocus amplifications you got with your PCRs.

Answer 8: As it is, we only validated the specificity of all primer pairs in silico by blast searches against

available tomato sequences curated at the NCBI database (Lines 157-159). 

Comment 9: Options for all programs used should be included in the methods. It would be very hard to repeat this work based on the details given.

Answer 9: Default settings were used for all the programs. (Line 147)

Comment 10: Line 171: Your sequencing coverage is pretty low for de novo assembly

Answer 10: For this study, we only requested for single lane sequencing. Multiple lane sequencing might

have helped us increase the coverage.

Comment 11: Line 178: You might have been able to adjust your mapping parameters to get better alignment, but since you don’t give these parameters it is hard to know.

Answer 11: Default settings were used to run the alignment. (Line 147)

Comment 12: Line 190: There is a PacBio assembly of LA2951 already (https://www.biorxiv.org/content/10.1101/2020.04.16.039636v1.full)

Answer 12: The assembly of LA2951 is available but there are existing provisions for the use of the assembly until the study is published (https://solgenomics.net/organism/Solanum_lycopersicoides/genome). As mentioned in response to Comment 2, this study is part of a bigger research that focuses on characterization of introgression lines derived from another S. lycopersicoides accession (LA1964) and we started working on that project way before the PacBio assembly of LA2951 was released. 

Comment 13: Line 194: In closely related species, a lot more of the genome is annotated as repetitive. Why do you think your estimate is so low?

Answer 13: We used a reference-guided assembly based on the tomato genome. Because of the genetic distance between tomato and S. lycopersicoides, only a moderate number of contigs assembled into longer scaffolds that made up the draft. Analysis of the repetitive region was based on that draft which explains the lower estimate we got (Line 211-213).

Comment 14: Line 201: I don’t understand why you say the assembly only captures less that 50% of the lycopersicoides genome. Your assembly length is actually a bit longer than the expected genome size. Are you basing this off of the % that mapped to Heinz? You should do Kmer analysis to see what the expected size is.

Answer 14: The de novo assembly does have a longer length but the final assembly that was used for all the structural analysis was scaffolded based on the tomato reference genome and not all contigs from the de novo assembly was used in the final assembly.

Comment 15: Line 231: Why would you do this manually? You have the mummer alignment and can generate a pipeline to do this.

Answer 15: The indel markers were designed based on BWA alignment even before we decided to analyze the S. lycopersicoides genome to generate markers that we need for a bigger project (see answers to Comment 2). The MUMmer alignment was performed at a later stage.

Comment 16: Line 253: I’m curious why you did not use bioinformatics approaches to find the primers in these other species first. This approach would have given you a more refined list for PCR testing.

Answer 16: This study is a part of a bigger project and the markers were basically designed specifically for S. lycopersicoides and tomato polymorphism. The transferability in other species was tested later on.

Comment 17: Figure1 is very low resolution in my copy

Answer 17: Please download the figure file from the link on the upper-right corner of the page containing the figure. The figures that come with the pdf generated from the submission is always blurred.

Comment 18: Figure 3: the label for chromosome 4 has shifted to a different line

Answer 18: The label for chromosome 4 in Figure 3 has been adjusted.

Comment 19: Do the markers work in other lycopersicoides accessions? You could check LA2951 which is a more finished assembly.

Answer 19: The markers have been tested in two other accessions of S. lycopersicoides, LA2951 and LA2387 (Table 4).

Comment 20: How did you use BWA to align Lyd to Heinz? Line 144

Answer 20: Tomato cv. Heinz was used as a reference genome and the ABySS contigs were aligned to the tomato reference using BWA.

Comment 21: How did you handle multiply mapped contigs or regions of the Heinz genome that had many contigs aligning?

Answer 21: PAGIT software was able to manage the multiple contigs mapped to generate a single consensus sequence.

Comment 22: The reads should be submitted to SRA and the markers to solgenomics.net

Answer 22: The raw reads have been submitted to SRA and assigned the accession number SRX9292807. Marker information including sequences, chromosome location, map position in the chromosome, annealing temperature, and motif are made available in the S1 and S2 Tables.

---

## [Editor Report · Decision Letter 1]

11 Nov 2020

Development of a PCR-based, genetic marker resource for the tomato-like nightshade relative, Solanum lycopersicoides using whole genome sequence analysis

PONE-D-20-20252R1

Dear Dr. Angeles-Shim,

We’re pleased to inform you that your manuscript has been judged scientifically suitable for publication and will be formally accepted for publication once it meets all outstanding technical requirements.

Kind regards,

David D Fang, Ph.D.

Academic Editor

PLOS ONE
---

## [Editor Report · Acceptance letter]

13 Nov 2020

PONE-D-20-20252R1 

Development of a PCR-based, genetic marker resource for the tomato-like nightshade relative, *Solanum lycopersicoides* using whole genome sequence analysis 

Dear Dr. Angeles-Shim:

I'm pleased to inform you that your manuscript has been deemed suitable for publication in PLOS ONE. Congratulations! Your manuscript is now with our production department. 

Kind regards, 

on behalf of

Dr. David D Fang 

Academic Editor

PLOS ONE